# The Formation of All-Silk Composites and Time–Temperature Superposition

**DOI:** 10.3390/ma16103804

**Published:** 2023-05-18

**Authors:** James A. King, Xin Zhang, Michael E. Ries

**Affiliations:** School of Physics and Astronomy, University of Leeds, Leeds LS2 9JT, UK; mmjki@leeds.ac.uk (J.A.K.); cm16xz@leeds.ac.uk (X.Z.)

**Keywords:** silk-based composites, time–temperature superposition, biomaterials

## Abstract

Extensive studies have been conducted on utilising natural fibres as reinforcement in composite production. All-polymer composites have attracted much attention because of their high strength, enhanced interfacial bonding and recyclability. Silks, as a group of natural animal fibres, possess superior properties, including biocompatibility, tunability and biodegradability. However, few review articles are found on all-silk composites, and they often lack comments on the tailoring of properties through controlling the volume fraction of the matrix. To better understand the fundamental basis of the formation of silk-based composites, this review will discuss the structure and properties of silk-based composites with a focus on employing the time–temperature superposition principle to reveal the corresponding kinetic requirements of the formation process. Additionally, a variety of applications derived from silk-based composites will be explored. The benefits and constraints of each application will be presented and discussed. This review paper will provide a useful overview of research on silk-based biomaterials.

## 1. Introduction

In the production of composites materials, the increase in awareness of building a sustainable future has stimulated the idea of replacing non-renewable petroleum-based polymers with bio-based constituents that have low or zero carbon emissions [1]. For example, substantial research can be found on incorporating cellulosic fibres such as flax, hemp, kenaf and bamboo as reinforcement to produce natural-fibre-reinforced polymer composites (NFRPCs) for applications in the automotive industry [2,3,4]. Researchers believe that these NFRPCs have the potential to significantly reduce the weight of vehicles by 30% and cut down the overall manufacturing cost by 20% [5,6,7]. Comparatively, silks, a group of protein-based fibres that have evolved for millions of years and have properties that often exceed man-made materials, have had limited studies as a reinforcement for composites in engineering applications [8].

For the use of these natural-fibre composites in engineering applications, studies have pointed out that NFRPCs present a few drawbacks: (i) surface incompatibility between natural fibres and polymer matrices results in poor interfacial bonding between the fibre and matrix phases and negatively impacts the physical and mechanical performance of the composites; (ii) difficulty separating the constituents during the recycling process leads to degraded performance after recycling [5,9,10]. To overcome these concerns, considerable research has focused on exploring the potential of self-reinforced composites (SRCs), also referred to as single/all-polymer composites, where both the reinforcement and matrix are composed of the same polymer [11]. In 1975, Capiati and Porter first developed all-polyethylene (all-PE) composites through partially melting the crystals within the PE fibres, providing a gradual change in the morphology between fibre and matrix and resulting in a competitive interfacial shear strength comparable to conventional glass fibre reinforced with epoxy resin composites [12]. Following the success of all-PE composites, research on SRCs began around the world. The most-reported synthetic-fibre-based SRCs are fabricated using polyolefin-based (polyethylene and/or polypropylene), polyester-based (polylactic acid and/or polyethylene terephthalate) or polyamide-based (nylon) fibres or tapes. The techniques involved in fabricating SRCs include thermal processing [13], hot compaction [14,15,16,17], cool drawing [18], etc. [19].

Following the concept of SRCs and sustainable design of composite materials, all-natural fibre composites were first introduced by Nishino et al., leading to the design of all-cellulose composites (ACCs) [20]. There are generally two methods to fabricate ACCs: (i) an impregnation method (two-step approach), where fibres are impregnated by cellulose solution; and (ii) a surface-selective dissolution method, where surfaces of fibres are partially dissolved. Undissolved inner fibre cores serve as the reinforcement phase, while the dissolved fibres serve as the matrix phase upon coagulation [21,22,23]. Subsequently, research has been carried out utilising various sources of cellulose to fabricate ACCs, including ramie [24], cotton [25,26,27], hemp [28,29], flax [30], microcrystalline cellulose [21,31] and regenerated cellulose fibres such as Lyocell and Bocell [23,32,33]. Table 1 gives details of the mechanical properties of NFRPCs and SRCs, including synthetic-fibre-based and natural-fibre-based examples. It is shown that all-natural-fibre composites display mechanical properties comparable to those of NFRPCs.

Great research effort has been put into exploiting natural-fibre-reinforced composites for biomedical applications, owing to their excellent mechanical properties, biocompatibility and tunability [52]. For example, owing to the specific hierarchical structure of silk protein, silk fibres exhibit superior mechanical, bio resorbable and biocompatible properties, resulting in their popular usages in biomaterial applications including tissue engineering, drug delivery and biodegradable medical devices [53,54,55,56,57,58,59,60,61]. A. Leal-Ega ña and T. Scheibel [62] reviewed silk-based materials for cell culture and tissue engineering, and they suggested that silk materials offer several advantages, including low toxicity, low immunoreactivity, wide pore-size distribution and elastic properties. They also pointed out that the mechanical properties of materials, including Young’s modulus, extensibility, strength and toughness, are crucial factors for both in vitro and in vivo biomedical applications, since these factors can greatly impact the cell behaviour, tissue regeneration and the integration of the graft into the body.

It is generally acknowledged that some key factors need to be considered when developing and selecting biocomposites for biomedical applications. For example, the chemical composition and surface biocompatibility of the chosen biocomposites must be able to provide a suitable chemical, biological and morphological environment for the host tissue. As addressed earlier, the chosen biocomposite’s mechanical properties need to be similar/comparable to the host tissue’s mechanical properties in order to prevent adverse reactions such as immunity response, allergic reactions and inflammation [8,63,64,65]. In order to optimise the properties of NFRCs for use in biomedical applications, researchers have conducted chemical and physical treatment of the natural fibres, while maintaining the requirements of biocompatibility, bioactivity and biostability. This has proven to be challenging because any additives comprised of different chemicals could potentially affect the biocompatibility and biodegradability of the resulting composite [52,66]. Hence, this opens up opportunities for exploring the potential for all-natural-fibre composites in biomedical applications, since homogeneity in the chemical-element composition of this type of composites can promote fibre/matrix adhesion and reduce complex synergistic immunoresponses.

Nevertheless, it is remarkable that, given the extraordinary physical and mechanical properties of silk proteins, more research investment has yet to be made with respect to the employment of silk fibres in producing silk-based biocomposites [42]. In this review, the structure of the fibrous proteins in silkworm silk and the properties of silk-based composites will be reviewed. Particularly, recent studies that employed several analysis methods to achieve the tunability of mechanical properties of silk biocomposites will be examined. Additionally, different applications of silk-based composites, their benefits and requirements, as well as a number of characterisation techniques will also be discussed. This is currently an evolving field with significant potential for producing silk-based composites with enhanced interfacial bonding, easier recyclability, and great tunability for future technological applications. It is believed that such an article would be of interest to the silk research world.

## 2. Silk Structure and Silk-Based Composites Properties

Silk is a common biological protein formed of a complex hierarchical structure with variable chemical compositions. The most common of these are the silk sericin and silk fibroin (SF) proteins. SF typically has a hexapeptide primary sequence dominated by glycine amino acid units, as seen in Figure 1 [67]. Raw fibre sheets of these biomaterials have inherent flaws compared to composites. Existing voids act as water channels to allow degradation by wetting, and hydrogen bonds can be broken by water molecules, which allows solvation and plasticisation [68]. This and other issues can be overcome by their inclusion in composites, which can improve material properties.

Silk forms a complex semi-crystalline hierarchical structure in its native state. The most common commercial silk is from the silk worm, *Bombyx mori*, but silk is also produced by spiders and other species [69]. It exists in heavy or light chains of 390 or 25 kDa, respectively, with chaperonin-like P25 proteins in a 6:6:1 ratio displaced along fibril axes [70,71]. Natural fibres of *Bombyx mori* are typically formed of two filaments of SF with a surrounding matrix of gummy silk sericin to maintain integrity. The continuous phase of silk sericin is removed by degumming using hot water, alkaline or acidic solutions, or other methods. The structure can be seen in Figure 2 [69]. In *Bombyx mori*, SF content is 66.5–73.5 wt%, while sericin content makes up 26.5–33.5 wt% [72].

The source of this silk can alter its properties with variations in strength, toughness and finish. Spiders alone can have seven different types of silk, controlled by the amino acid content of the protein. Dragline silk is rich in analine [73]. The precise effects of glycine and analine concentrations on SF crystallinity are disputed [69,73]. Shear and elongation stress also control conformation and crystallinity and, hence, material properties. Silk can reach a strength-to-weight ratio five times that of steel and three times that of Kevlar [73].

The high glycine content of SF allows for tight and stable packing of antiparallel β-sheet crystallites in SF. These are associated with the mechanical strength of silk [69]. In its native state, *Bombyx mori* SF has a very complex structure with differing amino acid sequences promoting high- and low-order structures [74]. This typically consists of 56±5%
β-sheet crystallites and 13±5%
α-helix conformations, with the remaining molecules disordered [75]. SF can form silk I, II, or III. Silk I has an α-helix or zigzag spatial conformation and is metastable. It often exists in phases within amorphous regions of semicrystalline SF and is shown to exhibit good swelling properties [69,76]. As with other crystal polymorphs, the structure has good chemical, thermal and enzymatic stability [76]. Silk II has the less soluble, stable β-sheet crystal structure, which is a monoclinic system [69]. Silk III is an unstable polymorph only seen at air–water interfaces in regenerated silk [67]. High silk II content is often associated with preferable material properties such as strength and toughness.

## 3. Published Methods of Tailoring Mechanical Properties of Silk Composites

Mechanical properties of NFRPCs are one of the most important characteristics for their use in engineering and biomedical fields, and it is essential for composite materials to provide adequate mechanical behaviour (tensile, flexural, impact and hardness) based on their desired applications [77,78]. The mechanical properties of NFRPCs depend on various parameters, such as fibre alignment, fibre length, fibre orientation, volume fraction of fibres, aspect ratio of fibres and fibre–matrix adhesion [79,80,81,82,83]. Studies investigating mechanical performance of NFRPCs have focused on two major aspects: (i) influences of various treatments of fibres (physical, chemical and biological), fibre content, fabrication process and external coupling agents on mechanical properties; and (ii) incorporating experimental data and well-established models to predict mechanical behaviour [84,85,86,87,88,89,90,91,92,93].

Natural silk fibres from *Bombyx mori* have relatively high mechanical properties: 300–740 MPa (ultimate strength), 4–26% (breaking strain), 10–17 GPa (Young’s modulus) and 70–78 MJm${}^{-}$${}^{3}$ (toughness). These properties often exceed those of synthetic fibres such as nylon, Kevlar and polypropylene [54,61,94,95,96,97,98]. The variation in mechanical properties of *Bombyx mori* silk fibres comes from several factors, such as the food, rearing conditions and health of silkworms [99], differences in the spinning process (natural spinning, forced spinning at a controlled drawing rate and modulated spinning in an electric field) [54,100,101] and genetic modification of the silk sequence [102,103]. For the purpose of broadening silk-based applications, silk fibres generally undergo the process of degumming, dissolution and regeneration and subsequently form the formats of sponges, hydrogels, films, mats and composites for versatile applications. Research suggests that these SF materials show good biocompatibility with a series of cell types and also promote the characteristics of adhesion, proliferation, growth and functionality [104].

In particular, research on tailoring the mechanical properties of silk-based composites also follows the two major aspects addressed earlier: (i) applying experimental treatments to tune the mechanical properties in order to achieve the desired structure–property–application relationship, as summarised in Table 2; and (ii) using analytical models to predict mechanical performance for further facilitation of composite material design and optimisation. For example, Jiang et al. [105] fabricated ultrathin multilayer SF films using a spin-assisted layer-by-layer assembly method with 8% *w*/*v* aqueous SF solution. These films exhibited outstanding tensile strength (∼100 MPa), Young’s modulus (6–8 GPa) and toughness (328 kJ m${}^{-3}$), comparable to the toughness of many conventional polymer composites (<100 kJ m${}^{-3}$). Burger et al. [106] fabricated SF/cellulose composites in hydrogel form, with the cellulose content varying between 0–17%. They reported that the compression properties of the hydrogels can be altered by the content of cellulose, with the compressive modulus varying between 53.3 and 70.5 kPa and compressive strength between 11.7 and 17 kPa. The biological properties of this composite hydrogel were investigated and showed good cell adhesion, cell growth and osteocyte differentiation of MC3T3-E1 cells. As a result, they proposed this composite hydrogel as a suitable scaffold for use in bone-tissue engineering. More recently, Yu et al. [107] found that the mechanical properties and drug-release behaviour of silk hydrogels can be tuned through ethylene glycol diglycidyl ether-assisted crosslinking of SF with carboxymethyl chitosan (CMCS). They investigated the SF/CMCS composite hydrogel at various mass ratios (1:0, 1:1, 1:2 and 0:1). The compressive fracture stress and strain values achieved were greatest at a 1:1 ratio (SF:CMCS): 140 kPa and 50%, respectively. The fracture energy reached a maximum value of 11.07 kJ m${}^{-3}$, with a compressive modulus of 22.24 kPa. They proposed that the addition of CMCS increases the molecular weight of SF and potentially induces the conformational transition of SF to increase the content of β-sheet structure, thus enhancing the strength and elasticity of the composite hydrogel. They further tested the sustained release rate of the SF/CMCS hydrogel using trypan blue as the model drug and noticed the rate was almost 35% higher than that of the SF hydrogel [107]. This composite hydrogel is suggested to be used as a biomedical carrier for sustained drug release.

Studies also focus on exploiting analytical models to predict and tailor the mechanical properties of silk-based composites. It is acknowledged that modelling can reveal the structure–property relationship within composites with little time and cost compared to obtaining results from a range of experiments [108]. Lin et al. [109] constructed SF and collagen type II (col-II) composites in the membrane format with varied SF:col-II ratios (5:5, 7:3, 9:1). They employed a nonlinear viscoelastic constitutive model to predict the creep behaviour and tensile properties of these composite membranes and compared it with the corresponding experimental data. From their obtained stress–strain curves, it was observed that the Young’s modulus values increased with decreasing col-II and, conversely, the ultimate strain and tensile strength decreased with decreasing col-II. Importantly, both the experimentally observed creep strain and tensile stress–strain curves were in good agreement with predictions. Additionally, a live/dead cell double-staining and cell proliferation assay demonstrated that these composite membranes display good cell proliferation ability and biocompatibility and further indicated that the SF/col-II composite membrane is a potential scaffold material for cartilage repair. Guan et al. [110] focused on utilising a finite element analysis (FEA) model to investigate the importance of fibre properties and inter-fibre bonding strength on the resulting mechanical properties of silk-based composites. They simulated the inter-fibre bonding strength through constructing a fibre network with fixed short beams of sericin, and strength was varied by modifying the cross-sectional area and length. By combining conclusions from microstructural images of the cocoon samples after tensile failure with simulation results, they proposed that good mechanical properties of fibres and effective inter-fibre bonding are crucial for making a tough and strong fibre composite. In recent studies, all-silk composites (ASCs) were fabricated using a surface-selective dissolution method by Zhang et al. [111]. They reported that individual *Bombyx mori* silk threads can be partially dissolved in the chosen solvent (1-ethyl-3-methylimidazolium acetate) for various times and at various temperatures and can be further coagulated in a methanol bath. Upon doing so, the undissolved inner thread core acts as the reinforcement phase, while the dissolved outer layers of silk thread coagulated and formed a matrix phase surrounding the inner fibre core. Through use of wide-angle X-ray azimuthal scans, the fraction of preferred oriented crystalline and randomly oriented crystalline within the ASCs fabricated for different times and at different temperatures was compared, and the average crystalline orientation ($P_{2}$) values were calculated. They further noticed that the $P_{2}$ values were linearly related to the coagulated volume fraction of the matrix ($V_{m}$). After conducting tensile tests on ASCs fabricated for various times and at various temperatures, they found that the Young’s modulus values decreased linearly with increases in dissolution time and more rapidly at increased dissolution temperature (Figure 3a). They proposed that the change in the Young’s modulus values followed a time–temperature superposition principle and further confirmed it through the employment of a shifting method in log space (Figure 3b), leading to the construction of a master curve for the Young’s modulus values at a chosen reference temperature (Figure 3c). Furthermore, they applied the well-established rule of mixture models to investigate the influence of reinforced fibre volume on the resulting Young’s modulus of the composite. Though some discrepancies between experimental and theoretical results existed, the experimental data suggested that the fabricated ASCs showed effective stress transfer between fibre–matrix up to a matrix fraction of ∼70% (Figure 3d). Consequently, they demonstrated the potential of ASCs and the possibility of altering their mechanical properties with controlled fabrication processes, dissolution times and temperatures.

## 4. Applications of Silk Composites

Although the fundamental research of silk composites is interesting due to the inherent complexity of the natural system and preparation conditions, they are often researched with intended applications in mind. Hence, research can either be approached as bottom-up, fundamental, blue-sky research or challenge-driven top-down research [112].

Silks have been primarily used for biomedical applications as they are perceived to be biocompatible, biodegradable and non-toxic [113]. It is of note that biocompatibility is not universal and must be specific to tissue and wound to encourage the correct immune response during healing [112]. Current uses of silk include sutures, surgical meshes and medical fabrics. Coating silk fibres with regenerated SF for use as sutures is one of the first examples of ASC use in medicine. Some future applications still being developed include tissue engineering and wound healing [112]. Microneedles also offer an exciting new development in transdermal vaccine delivery. SF microneedles offer a solution for controlled-release drug delivery with minimally invasive techniques [114,115]. As shown in works by Tsioris et al. and Stinson et al., SF microneedles provide favourable mechanical properties, biocompatibility, biodegradability, benign processing conditions and the ability to maintain the activity of biological compounds in its matrix [114,115]. This biomaterial could offer a new application for engineered biocomposites of SF in which the techniques mentioned above may confer improved toughness over simpler SF microneedle arrays.

Scaffolds of biomimetic materials are common forms of biosynergistic composites and function as host environments for cell and tissue growth and proliferation [116]. In order to be biomimetic, these engineered tissues must regulate healing phases by imitation of immunoresponse signals [117]. When preparing a composite for tissue engineering, it must provide [118,119,120,121]:Appropriate surface roughness [118,120];Appropriate permeability and absorption [119];Correct release behaviour [119];Porosity [118,121];Structural stability [118];Appropriate mechanical strength [120,121];Thermal stability [118,121];Biocompatibility [118,120,121];Biodegradability [118,121].

This complexity requires careful preparation of aero- or hydrogels to meet these requirements. This, then, requires engineered tissues to mimic highly variable native or artificial tissue mechanical (shear, tensile or compressive) moduli [122,123,124]:Articular cartilage—0.4–1.6 MPa [122];Native femoral artery—≈9.0 MPa [122];Human medial meniscus—≈1.0 MPa [122];Fixation plates—≈700 MPa [123];Cancellous bone—0.05–5 GPa [124].

SF is confirmed to promote cellular adhesion and proliferation of fibroblasts and keratinocytes [121,125,126]. Another benefit for biomedical uses and tissue engineering is that these materials can degrade in vivo and trigger minimal inflammatory response. They also aid in cell growth due to the intrinsic biocompatibility of SF materials [127,128]. This eliminates the need for implant-removal surgery but requires understanding of the degradation process to ensure it does not compromise efficiency [69,127,129]. Silk-based composites have also been prepared with drug retention and release capabilities that show some promise for targeted-drug-release applications [130]. Utilising ASCs could provide samples with high homogeneity, good material properties and tunable biological interactions for continued use in these applications. It is essential in biomedical applications of these composites to mimic the natural environment: for example, in the proliferation of fibroblasts on the surface of breast implants [126,131].

Wahab et al. utilised silk’s heavy-metal-adsorption properties in conjunction with low-cost bentonite clay to produce a composite capable of adsorbing lethal heavy-metal ions from aqueous solution [132]. They successfully impregnated SF with bentonite clay and achieved high monolayer adsorption values for Cd(II), Pb(II), Hg(II) and Cr(VI) [132]. Silk composites could offer future solutions in water sanitation aids as an environmentally sustainable tool.

Spider silk is seen to be of such high tensile quality, even compared to *Bombyx mori* silk [112], that composites reinforced with spider silk components have been thought of for use in aerospace engineering [133]. Mayank et al. prepared a transparent epoxy/spider-silk composite with material strength within the safety margins of typical standard acrylic aviation windowpanes [133]. They also showed improved impact deflection and stress loading. As a weight-reduced non-magnetic material, silk composites could offer an innovative material option for the aerospace industry [133]. It is of note that improved interfacial interactions of ASCs are seen due to chemically identical components of the matrix and the reinforcing fibres [50]. This is due to a favourable energetic interaction developed in identical biopolymers compared to phase-separated interfaces of polymer blends [134]. This interface is least influential in miscible polymer blends with favourable interactions, such as those between cellulose and silk fibroin [134]. ASCs could, therefore, reduce the likelihood of the primary failure mechanism of composites: failure at the interface of the reinforcement and the matrix. This implies the viability of ASCs in the uses mentioned throughout this article

Lastly, silk composites have also been used for decorative and restorative challenges, in line with their historic use. Cianci et al. produced a hybrid SF/cellulose solution to support and recover aged silk fabrics [71]. This forms a silk composite in which the cellulose improves silk crystallinity and material properties, allowing the sustainment of aged and degraded silk fabrics [71]. Untreated pristine silk and the same fabric treated with 1.35/1.35 wt% SF/cellulose nanocrystals dispersed in water then dried gave axial forces at breaking of ≈30 N and ≈35 N, respectively. This was a marked improvement over silk treated with single-component dispersions [71]. Other hybrid material composites have been produced for structural applications. Kimura and Aoki reinforced plywood and medium-density fibreboard with silk fabric using a polybutylene succinate matrix to create decorative laminates with improved flexibility and impact resistance [135]. ASCs could be used in a similar decorative capacity with dyed patterns from the fabric retained in the final product. As shown by Cianci et al., this could offer a use for waste silk with tangible benefits for the end product [71]. Though these improvements imply an impactful contribution from silk within these composites, it is essential to evaluate sustainability in the creation of these products. If overused, silk could contribute to environmental burdens rather than reducing them by increasing water usage and transport costs. We suggest that focusing on reusing old silk and effectively disposing of waste samples will be of huge value going forward.

## 5. Conclusions

With recent awareness of global environmental challenges, it has become of increasing importance to utilise sustainable biomaterials to replace non-renewable alternatives. In this role, silk possesses unique qualities of strength and biocompatibility that highlight it as a key contributor to future developments in NFRPCs. Through this review, we discussed the behaviours and uses of silk-based composites and reported on the enhancement of mechanical properties of silk-based composites. We separated enhancements into experimental (chemical and/or physical) treatments to achieve desired properties and the use of analytical models to predict the mechanical properties for further design and optimisation of the structure and properties. As an example of this, we reported on the modelling of the fabrication of ASCs using time–temperature superposition. This can be used to manipulate the mechanical properties and morphology of composites via changes to the dissolution time and temperature that alter the volume fraction of the matrix. This gives rise to clearly defined trends of material properties as a function of time, temperature or matrix fraction. ASCs can therefore be manipulated for desired purposes with good interfacial bonding between the fibre and the matrix due to chemical homogeneity. Finally, we discussed the applications of silk composites. This review focused on biomedical uses, with other examples in water sanitation, aeronautical engineering and decorative functional materials. This shows a future of applications in many industries with challenges highlighted in upscaling as well as achieving true sustainability with widespread use of these new materials.

## Figures and Tables

**Figure 1 materials-16-03804-f001:**
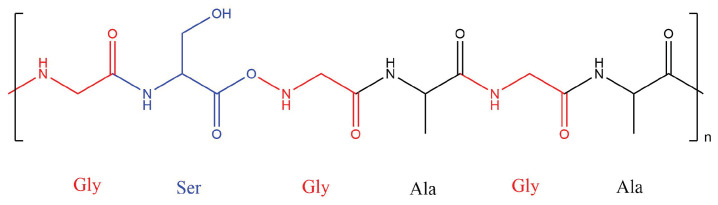
Illustration of the common chemical structure and amino acid sequence of a silk fibroin protein with a hexapeptide sequence.

**Figure 2 materials-16-03804-f002:**
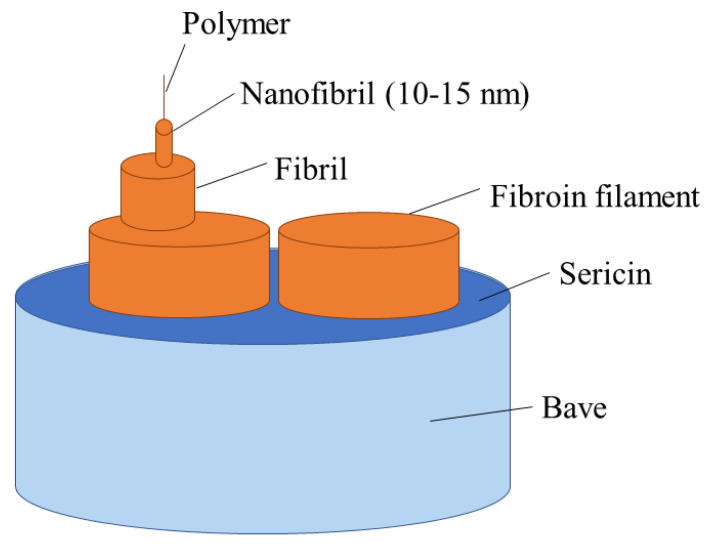
Illustration showing hierarchical structure of raw silk fibre with SF core and sericin coating.

**Figure 3 materials-16-03804-f003:**
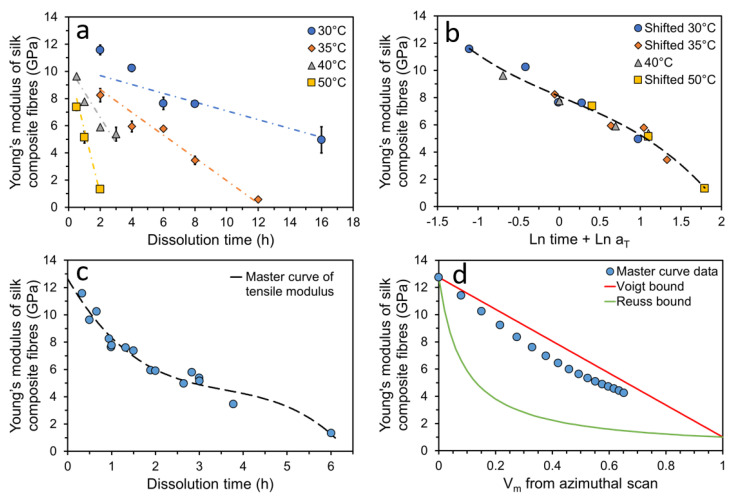
Tensile properties of all-silk composites in silk-thread format: (**a**) Young’s modulus of ASCs for various dissolution times and temperatures; (**b**) shifting each temperature dataset to a reference temperature dataset, 40 ${}^{\circ }$C, in log space; (**c**) master curve of Young’s modulus represented at 40 ${}^{\circ }$C; (**d**) Rule of mixture models predicting effectiveness of stress transfer between fibre–matrix. Reprinted (adapted) with permission from [111]. Copyright 2021 American Chemical Society.

**Table 1 materials-16-03804-t001:** Overview of the mechanical properties of NFRPCs and all-natural-fibre composites.

Composite Type ^1^	Tensile Strength (MPa)	Young’s Modulus (GPa)	Flexural Strength (MPa)	Flexural Modulus (GPa)	Reference
UD flax–epoxy	284	26	218	18	[34]
woven flax–epoxy	68.3	5.9	101.6	2.1	[35]
UD flax–PLA	252.3	14.9	-	-	[36]
Kenaf–HDPE	32–52	3.0–6.5	58–75	3.7–4.1	[37]
bamboo fibre–epoxy	48–119	-	107–162	-	[38]
wood pulp sheets–PLA	121	10.5	-	-	[39]
woven silk–epoxy	100–175	5–8	125–250	5–7	[40,41,42]
silk–PLA	71	4	97	4	[43]
all–PP	60–135	2.7–3.9	-	-	[44]
all–PLA	58–83	2–3.4	-	-	[45]
all–PLA	58–83	2–3.4	-	-	[45]
all–PET	225–350	8–10	-	-	[46]
all–PE	160	4.2	-	-	[47]
all–PA	80–192	-	-	-	[48]
all–cellulose	250–910	9–26	-	-	[49]
woven all-cellulose	40–55	1.5–4	-	-	[50]
all-silk	83–151	2.8–3.1	-	-	[51]

^1^ UD: unidirectional; PLA: polylactic acid; HDPE: high-density polyethylene; PP: polypropylene; PET: polyethylene terephthalate; PE: polyethylene; PA: polyamide.

**Table 2 materials-16-03804-t002:** Summary of the tailored mechanical properties of silk-based composites through various experimental treatments.

Formats of Silk-Based Composites	Experimental Treatments	Mechanical Properties	Biological Properties	Proposed Application	Reference
SF films	Spin-assisted layer-by-layer assembly method with 8% *w*/*v* aqueous SF solution	Tensile strength ∼100 MPa, Young’s modulus 6–8 GPa, Toughness 328 kJ m${}^{-3}$	-	Microscale biodevices, synthetic coatings for artificial skin	[105]
SF/cellulose composite hydrogel	Varying the content of cellulose between 0–17%	Compressive modulus 53.3–70.5 kPa, compressive strength 11.7–17 kPa	Good cell adhesion, cell growth and osteocyte differentiation of MC3T3-E1 cells	A suitable scaffold for use in bone-tissue engineering	[106]
SF/CMCS composite hydrogel	Ethylene glycol diglycidyl ether-assisted crosslinking of SF with CMCS and various mass ratios of SF/CMCS (1:0, 1:1, 1:2, 0:1)	When SF:CMCS is 1:1, compressive fracture stress and strain is 140 kPa, 50%, fracture energy 11.07 kJ m${}^{-3}$, compressive modulus 22.24 kPa	High sustained release rate when using trypan blue as the model drug	Biomedical carrier for sustained drug release	[107]

SF: silk fibroin; CMCS: carboxymethyl chitosan; MC3T3-E1 cells: osteoblastic cell line derived from mouse calvaria.

## Data Availability

Data and sources available upon request from authors.

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
