# Peer review of "The Formation of All-Silk Composites and Time–Temperature Superposition"

_materials, 2023, doi:10.3390/ma16103804_

Round 1

Reviewer 1 Report

1. As for Table 1, This table summarizes the mechanical properties of natural fiber reinforced polymer composites. It would be better if there were more tables of this type.

2. In page 4 line 129-130, the authors said “It is most soluble and often exists in phases within amorphous or solvated regions”. However, recent studies have shown that silk fibroin materials with Silk I crystal structure have good chemical, thermal, time, and enzymatic stability, and exhibit good swelling rather than solubility properties. (1.Zhao, M.; Qi, Z.; Tao, X.; et al. Chemical, Thermal, Time, and Enzymatic Stability of Silk Materials with Silk I Structure. Int. J. Mol. Sci. 2021, 22, 4136.https://doi.org/10.3390/ijms22084136).

3.In addition to the material forms mentioned in the article, silk fibroin can also be prepared into microneedles for drug delivery. Here are some of the latest papers on silk fibroin microneedles.

(1. Konstantinos, Tsioris, Waseem, et al. Fabrication of Silk Microneedles for Controlled-Release Drug Delivery[J]. Advanced Functional Materials, 2011, 22(2):330-335ï¼› 2. Jordan, A, Stinson, et al. Silk Fibroin Microneedles for Transdermal Vaccine Delivery. Acs Biomaterials Science & Engineering, 2017.)

4. It is the best for the authors to add some pictures (focusing on the work introduced in the review) and summary tables, which will help readers read and enhance the readability of this review.

Reviewer 2 Report

The authors of this work show the properties of silk fibroin and the contribution of these properties in different kinds of composites, especially where polymeric matrices are used.

The information and structure are adequate, however, in order to support how silk fibroin provides added value to the properties of the composites, the following information must be added:

1. A section where it is explained how the SF chemically interacts with the polymeric matrices to grant the improvement of the mechanical properties, and in which media there is a better interaction, and explain how SF improves the biocompatibility of polymer matrices.

2. The author's own point of view should be given about what is described and not be limited only to what is described in the bibliography.

3. Add the following bibliography:

https://doi.org/10.3390/insects13030286

https://doi.org/10.3390/polym11030451

https://doi.org/10.1002/jbm.a.37285

https://doi.org/10.1002/jbm.b.33973

Reviewer 3 Report

This review is very interesting and appropriate for publication. However, please check the text throughout the manuscript. It is also found several typing errors in the manuscripts.

It is just a minor mistake in English. Some wording is wrong typing in the text.

Reviewer 4 Report

This review article provides substantive value related to the development of sustainable composites and  an overall idea of the state-of-art in all sustainable silk composites.

The authors analyzed the extensive literature concerning both the theory and practice related to the production and mechanical characterization and applications  of silk composites.

This review article is well written and properly organized.

I will recommend this article for publication.

Reviewer 5 Report

I will show you some comments on the review paper “The formation of all silk composites and time-temperature superposition”.

I think that this article may publish as it is, but the article is getting better when authors perform minor reversions as follows;

Comments:

In section 3, there are lots of the mechanical data such as Young’s modulus, the ultimate strength and the toughness of silk composites from line#145 to line#185. They should be listed as Table 2 because it is more convenient for the readers of this article.

In section 4, every particular item from line#251 to line#259 should correspond to reference numbers [i.e. 115-118 and more] such as line#263 to line#267.
